# Electrodeposition and Corrosion Properties of Nickel–Graphene Oxide Composite Coatings

**DOI:** 10.3390/ma14195624

**Published:** 2021-09-27

**Authors:** Vitaly Tseluikin, Asel Dzhumieva, Andrey Yakovlev, Anton Mostovoy, Svetlana Zakirova, Anastasia Strilets, Marina Lopukhova

**Affiliations:** Engels Technological Institute, Yuri Gagarin State Technical University of Saratov, Polytechnichskaya St., 77, 410054 Saratov, Russia; aselka2796@gmail.com (A.D.); aw_71@mail.ru (A.Y.); Mostovoy19@rambler.ru (A.M.); zakirova-71@mail.ru (S.Z.); buran-92@mail.ru (A.S.); mlopuhova@yandex.ru (M.L.)

**Keywords:** composite electrochemical coatings, nickel, grapheme oxide, structure, corrosion properties

## Abstract

Nickel-based composite electrochemical coatings (CEC) modified with multilayer graphene oxide (GO) were obtained from a sulfate-chloride electrolyte in the reverse electrolysis mode. The microstructure of these CECs was investigated by X-ray phase analysis and scanning electron microscopy. The corrosion-electrochemical behavior of nickel–GO composite coatings in a 0.5 M solution of H_2_SO_4_was studied. Tests in a 3.5% NaCl solution showed that the inclusion of GO particles into the composition of electrolytic nickel deposits makes their corrosion rate 1.40–1.50 times less.

## 1. Introduction

Composite electrochemical coatings (CECs) are obtained by the co-deposition of metals with various dispersed particles from electrolyte suspensions [1,2,3]. Once incorporated into the metal matrix, the particles improve the operational properties of galvanic deposits (wear resistance, hardness, corrosion resistance, etc.). Therefore, CECs are used in various industries, and the development of new composite coatings is an important scientific and technical task. Among CECs, coatings based on nickel [4,5,6,7,8,9,10,11] and its alloys [12,13,14,15,16,17,18,19] have become widespread, which is due to the ability of nickel to form electrolytic deposits with dispersed particles of a different nature which have good adhesion to the metal substrate. Nickel-based composite coatings are characterized by hardness and resistance in various corrosive environments. They are used for machine parts and mechanisms operating in hard, as well as especially severe conditions [1,2].

The characteristics of the CECs are largely determined by the properties of the dispersed phase. Currently, a significant number of studies are devoted to the electrodeposition of composite nickel coatings modified with nanosized and nanostructured carbon materials: fullerene C_60_ [4], carbon nanotubes [5,6,7], nanodiamonds [10], carbides [8,11], etc. Among the carbon compounds, graphite and its derivatives are of interest. A distinctive feature of graphite is its pronounced layered structure. The graphite layer can act as a donor by reacting with various oxidants. Thus, when graphite interacts with strong inorganic acids, graphene oxide (GO) is formed, which is graphene layers chemically bonded to oxygen-containing functional groups (hydroxyl, epoxy, carbonyl, etc.) [20]. Graphene and graphene oxide are being extensively explored due to their remarkable operational properties (physicomechanical, electrical, thermal, etc.) [20].

Previously, we studied nickel coatings modified with fullerene C_60_ [4] and carbon nanotubes [15,19]. The purpose of this work is to obtain nickel–GO CECs in the reverse electrolysis mode, to investigate their structure and corrosion properties.

The advantage of unsteady electrolysis consists in a much larger number of parameters that control the process of coating deposition [21,22]. The use of a reverse current makes it possible to increase the content of the dispersed phase in the CEC structure and to achieve its uniform distribution over the thickness of the deposit. During the anodic period, a certain number of particles are released and remain in the near-electrode layer. In the subsequent cathodic period, these particles are included in the electrolytic deposit, increasing the content of the dispersed phase in it [22]. As a result, coatings with improved operational properties are formed.

## 2. Materials and Methods

Composite nickel–GO coatings were deposited on a steel base (steel 45) from the electrolyte, the composition of which is shown in Table 1.

Multilayer graphene oxide was added into the electrolyte bath as a powder with a particle size not exceeding 10 μm. The process of CEC deposition was carried out with constant stirring of the electrolyte. Electrochemical deposition of pure nickel was obtained from the above solution without the dispersed phase of GO. The thickness of studied coatings was 20 μm.

Multilayer graphene oxide was synthesized electrochemically in a galvanostatic mode by anodic oxidation of natural graphite powder GB/T 3518-95 (Sunshine Resources Holdings Limited, Beijing, China) with a capacity of 700 Ah/kg 83% H_2_SO_4_ (high purity grade) served as the electrolyte. A detailed description of the method for the synthesis of GO and the composition of the resulting products are presented in [20].

The specific surface area of graphene oxide was determined by the Brunauer–Emmett–Teller (BET) method using a NOVA 2000e analyzer (Quantachrome Instruments, Boynton Beach, FL, USA).

Structural studies were carried out using a scanning electron microscope with a built-in EXplorer energy dispersive analyzer (Aspex, Delmont, PA, USA). X-ray phase analysis (XRD) was performed on an ARL X’TRA X-ray diffractometer (Thermo Fisher Scientific (Ecublens) SARL, Ecublens, Switzerland).

Electrochemical measurements were performed on a P-30J pulse potentiostat (Elins, Moscow, Russia). The potentials were set relative to saturated silver chloride reference electrode and recalculated using a standard hydrogen electrode (SHE).

The electrodeposition of nickel-based coatings was studied in the reverse mode at cathode current density i_c_ = 10 A/dm^2^ and anode current density i_a_ = 1.5 A/dm^2^. The ratios of the cathodic period (t_c_) and anodic period (t_a_) were 10: 1 s, 12: 1 s, 14: 1 s, 16: 1 s.

The corrosion-electrochemical behavior of nickel and nickel–GO coatings was investigated by the potentiodynamic method in a 0.5 M H_2_SO_4_ solution (potential sweep rate Vp = 8 mV/s). To determine the corrosion rate, the coated samples were tested in a 3.5% NaCl solution.

## 3. Results and Discussion

Dispersed particles of all kinds can easily co-deposit with nickel, but their use gives different effects [1,2]. The inclusion of the dispersion of graphene oxide into the composition of the sulfate-chloride electrolyte of nickel plating greatly affects the kinetics of the electrode processes. A decrease in potential jumps during the transition from the cathodic to the anodic period can be seen on the E, t-curves (Figure 1). During the deposition of the nickel–GO CEC, a shift of potentials towards more electronegative values is observed as compared to nickel without a dispersed phase. When the deposition curve of the composite coating is shifted in the negative direction compared to the curve of the individual metal, superpolarization occurs. In the opposite case, they talk about depolarization. Hence, in the presence of graphene oxide, the cathodic process proceeds with superpolarization.

A study by scanning electron microscopy (SEM) made it possible to find that the graphene oxide has a layered structure with a developed surface (Figure 2a,b). The specific surface area of the GO, determined by the Brunauer–Emmett–Teller (BET) method, was 46.78 m^2^/g. Probably, adsorption of cations from the electrolyte solution can occur on the graphene oxide, which leads to the formation of a positive charge of the dispersed phase particles. Therefore, the transfer of GO to the cathode occurs not only due to convection, but also under the action of electrophoretic forces. The adsorbed ions are likely to participate in the “bridging” binding of the dispersed phase with the electrode surface [23]. This binding weakens the wedging pressure of the liquid layer between the GO particles and the cathode, thus enhancing adhesion.

According to the results of the XRD analysis of the nickel–GO coating sample, peaks corresponding to the phases of nickel and carbon in nickel were revealed (Figure 3). Those peaks of Ni will change with the existence of GO. SEM images (Figure 4) clearly show that the surface microtopography changed during the transition from a nickel deposit without a dispersed phase to a nickel–GO CEC. The composite coating had an ordered cellular structure (Figure 4b), while the structure of pure nickel was close to the X-ray amorphous one (Figure 4a). Obviously, graphene oxide particles act as crystallization centers and contribute to the distribution of metallic nickel over the cathode surface. It should also be noted that the nickel–GO CECs were dense and uniform.

Corrosion resistance is an important operational property of electrolytic deposits. It follows from the anodic potentiodynamic curves (PDC) of nickel and nickel–GO CECs (Figure 5) that the particles of the dispersed phase increased the potential and, accordingly, decreased the current of active anodic dissolution of the studied coatings. The corrosive behavior of the composite coatings was largely due to the properties of the metal matrix; therefore, the potentials of the onset of passivation of pure nickel and the nickel–GO CEC are close. A characteristic feature of the anodic PDC of the nickel–GO CEC is a noticeable broadening of the passive range, while for a nickel coating without a dispersed phase, it is smeared out. In the far anodic potential range, GO particles in the bulk of the nickel matrix also have a significant effect on the PDC behavior.

The corrosion rate of the coatings under study was determined by the weight loss when kept in 3.5% NaCl for 24 h (the samples were weighed before and after immersion) using the following formula [24,25]:
(1)$$\mathrm{Corosion}\;\mathrm{rate}=\frac{K\;W}{A\;T\;D}$$
where K is a constant (8.76∙10^4^), W is mass loss in g, A is the exposed area of the nickel and nickel–GO coating sample (1 cm^2^), T is the immersion time in h, and D is the density of nickel (8.90 g/cm^3^).

Corrosion tests showed that the corrosion rate of the nickel deposits without a dispersed phase is 1.40–1.50 times higher than that of the composite nickel–GO coatings (Table 2).

The revealed effect may be due to several factors. When nickel is deposited from sulfate-chloride electrolytes, matt porous coatings are formed [1,2]. In the process of the inclusion of graphene oxide particles into the nickel matrix, the pores overlap. The bigger the overlap (coverage) area of the surface by particles of the dispersed phase, the more resistant to corrosion the composite coatings are, because this ensures a uniform distribution of the corrosive current between the centers that prevent its propagation. Moreover, the effect of the dispersed phase in the CEC structure on corrosion is observed only when more corrosion-resistant matrixes than metal ones are formed by particles at phase boundaries or throughout the entire volume of compounds. Otherwise, the development of the corrosion process will not stop, but will bypass the particle. Such compounds are obviously formed in the case of composite nickel–GO coatings.

## 4. Conclusions

Based on the conducted studies, we can draw the conclusion that the addition of a dispersion of multilayer graphene oxide into the sulfate-chloride nickel-plating electrolyte results in the formation of CECs. The inclusion of GO particles into the composition of nickel deposits leads to a change in the microstructure of their surface. Graphene oxide has a decisive effect on the corrosion properties of the studied composite coatings. The inclusion of GO particles into the composition of electrolytic nickel deposits makes their corrosion rate 1.40–1.50 times less. Nickel–GO CECs can be used as corrosion-resistant coatings in mechanical engineering and the chemical industry.

## Figures and Tables

**Figure 1 materials-14-05624-f001:**
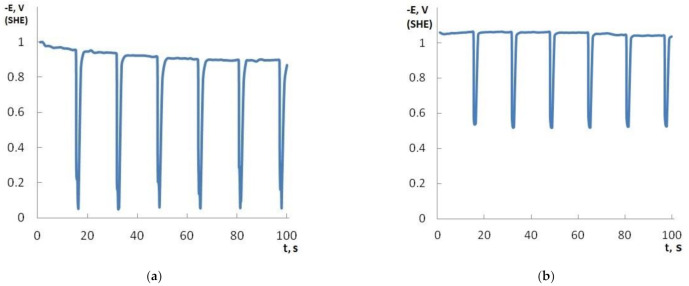
E, t-curves of electrodeposition of nickel (**a**) and nickel–GO CEC (**b**) at the time ratio of the cathodic and anodic periods t_c_/t_a_ = 16:1 s.

**Figure 2 materials-14-05624-f002:**
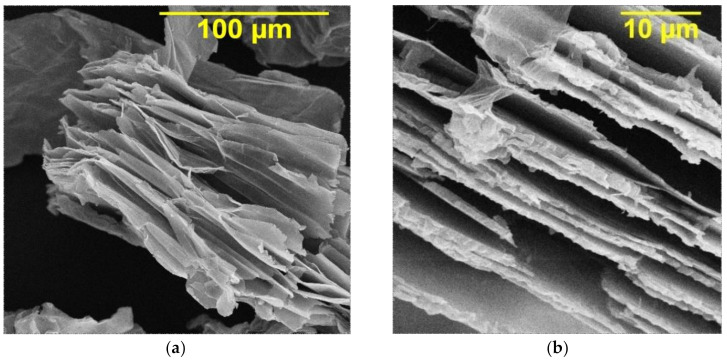
SEM images of the graphene oxide structure with various magnifications: ×1000 (**a**) and ×5000 (**b**).

**Figure 3 materials-14-05624-f003:**
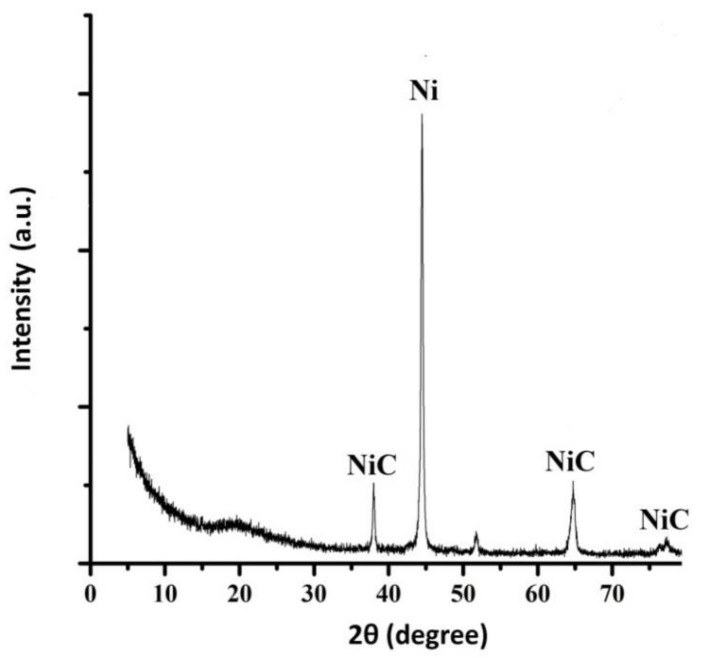
X-ray diffraction pattern of the nickel–GO CEC obtained at the time ratio of the cathodic and anodic periods t_c_/t_a_ = 10:1.

**Figure 4 materials-14-05624-f004:**
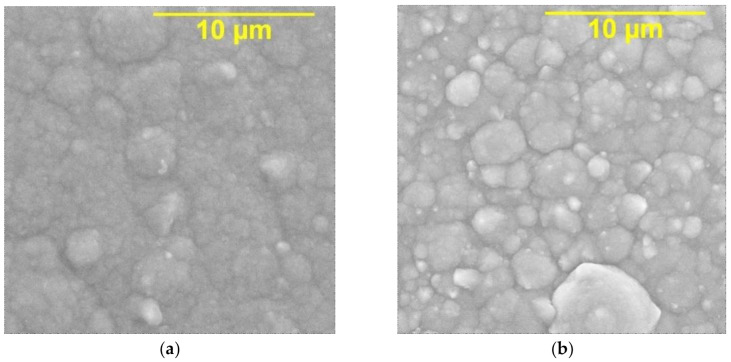
SEM images of nickel surface (**a**) and nickel–GO CEC (**b**). The time ratio of the cathodic and anodic periods t_c_t_a_ = 10:1. Magnification ×10,000.

**Figure 5 materials-14-05624-f005:**
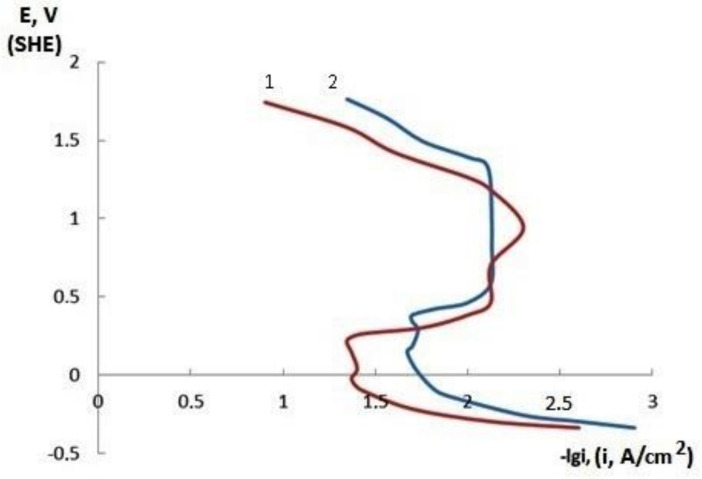
Potentiodynamic polarization curves of nickel (1) and nickel–GO CEC (2) in 0.5 M H_2_SO_4_ (coatings were obtained at the time ratio of the cathodic and anodic periods t_c_/t_a_ = 16:1).

**Table 1 materials-14-05624-t001:** Electrolyte bath composition and deposition parameters used for nickel–GO composite coatings.

No.	Electrolyte Composition	Concentration, g/L	Deposition Parameters
1	NiSO_4_·7H_2_O	220	Temperature t = 45 °C
2	NiCl_2_·6H_2_O	40	–
3	CH_3_COONa	30	–
4	Graphene oxide	10	–

**Table 2 materials-14-05624-t002:** Corrosion rate of nickel-based coatings, mm/y.

Time Ratio t_c_/t_a_, s	Nickel	nickel–GO CECs
10:1	0.656	0.451
12:1 14:1 16:1	0.533 0.410 0.246	0.369 0.287 0.164

## Data Availability

Not applicable.

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
