# Peer review of "Electrodeposition and Corrosion Properties of Nickel–Graphene Oxide Composite Coatings"

_materials, 2021, doi:10.3390/ma14195624_

Round 1
Reviewer 1 Report
This is a short manuscript which compares the corrosion resistance of one nickel and one nickel-GO coating. The authors report that independently of the ratio of cathodic and anodic period, the corrosion rate is about 1.5x lower for the latter sample.
(1) My main impression is that the subject of the manuscript is really, really restricted. The single composition investigated sounds random, there is no dependence of the corrosion rate or other properties - see also the next comment - on the GO content. In my opinion it is very necessary to support the composition investigated. To be constructive: was it optimized previously with respect to some other quantity than corrosion rate? Or does it correspond to optimized content of carbon in some other form?
(2) The presented motivation is that "Nickel-based composite coatings are characterized by hardness ... They are used for machine parts and mechanisms ...". I am wondering whether the incorporation of GO improves the corrosion resistance at preserved values of other functional properties (to put an example: resistance to cracking) or whether it affects other properties as well.
(3) Last remark concerning the restricted subject: because the authors investigated a single composition, the plural "coatings" or "CECs" is confusing for the reader.
(4) The caption of Figure 2 "Magnification x500 (a), x5000 (b)" is in contradicion with the scale bars inside the figure (for the figure size in the .pdf which I got) ... the magnification seems to be ~x140 and ~x1400.
(5) I suggest to spell out tc/ta in the figure captions ... the present captions are not understandable without reading the text.
(6) Figure 2a in the present manuscript and Figure 2a in author's recent paper in Crystals (doi.org/10.3390/cryst11040415) are obviously the same. The overlap in itself is probably OK, but the authors should duly cite the aforementioned paper.
Author Response
Authors are sincerely grateful to Reviewer for a detailed analysis of the work and valuable comments. The answers to the comments are in the attached file.

Reviewer 2 Report
The manuscript describes the preparation of Nickel - Graphene Oxide Composite Coatings by electrodeposition and analyzes the Corrosion Properties. It is interesting and well organized, furthermore could be published after minor revision.
- The authors have previously published other manuscripts about this topic, see ref 3, 4, 15, etc. However, it is not clearly in the introduction and conclusions the Highlights of this work and the important new results described in this manuscript, furthermore they didn’t compare these results with the previous published results. The conclusions are concise, and the possible applications of these coatings are not describe.
- It is well kwon that the XRD diffractogram of GO is usually characterized by a characteristic peak at 2θ=10°, but the XRD diffractogram in figure 3 don’t revel this characteristic peak. The authors must improve this analysis and discussion. They must verify the experimental result and if necessary, explain the absence of GO peak.
- The authors must improve the SEM characterization of Nickel-Graphene Oxide Composite to better understand the morphology of composite. The magnification of images of Figure 4 didn’t allow to verify the morphology of sample.
- The authors must add information about the grade of oxidation of GO as precursor and in addition they must verify the possible reduction process of GO during the electrodeposition and after the preparation of Nickel-graphene oxide composite.
Author Response

(The authors gave the same response as above.)

Round 2
Reviewer 1 Report
I confirm that the authors addressed my small technical comments.
The first two comments of mine, pointing out the very restriced subject, are answered by "this is our first paper dedicated to nickel-GO coatings ... we will continue working in this direction ... will study the cracking resistance and other physical and mechanical properties". If I was the authors, I would wait for more results before the publication ... but I am leaving this on the editor to decide.
Reviewer 2 Report
Accept in present form
This manuscript is a resubmission of an earlier submission. The following is a list of the peer review reports and author responses from that submission.
Round 1
Reviewer 1 Report
Manuscript entitled „ Electrochemical Deposition and Corrosion Properties of Nick[1]3 el-Graphene Oxide Composite Coatings “ was submitted to be considered for publication on Nanomaterials. The authors presented the electrochemical deposition and corrosion testing on Ni-GO composite coating. The idea of dispersing GO into the electrolyte to deposit Ni is not new. There are a lot of the available studies on this topic, which show not only the surface analysis, but also mechanical performance, and so on. In contrast, the presented data as well as the discussion are not sufficient and precise to be able to publish on Nanomaterials. Thus, it is not recommended in its current state.
Details are followed:
- Introduction
The authors did not persuasively give the reason why they use Ni and GO in their coating. The benefit of these materials were describe with nonspecific and not scientifical terms such as “different nature” (line 28).
- Materials and methods.
An explanation with scheme and details of the reverse electrodeposition method are needed.
- Results and discussion
Line 82-89: the authors discussion about the effect of GO on thze voltage of electrodeposition and came to the conclusion of “superpolarization” without further explanation, what they mean with this term.
Line 94, they mentioned that “graphene oxide has a layered structure with a sufficiently developed surface” -> how do I know that it is sufficient or not? What is the parameter? BET result is also not especially high. it seems that the GO was not really exfoliated.
The discussion in line 96-103 is a speculation. There are no data to support this suggestion.
XRD data is not properly and precisely analyzed. The diffraction pattern should be analyzed using kind of standard database, then matching process to find out which phase the Ni exist. Like Ni 200 (about 52.5), Ni 111 around 45, Ni 220 around 76.8 and not NiC!!! Those peaks of Ni will be change upon GO existence, but not because of NiC. This is realy wrong.
SEM image quality is so low that it did not clearly show the difference between two surfaces. Based on these images, GO seems to be agglomerated.
Figure 5 is not properly presented. The curve also cannot show the repassivation potentials because it is not a cyclic voltammetry.
The corrosion rate cannot be in the unit of mm/g, normally it should be mm/year. It is anyway, no practical meaning to test with only 24h of immersion. The rate is also pretty high, in all cases, that I don’t understand how one can use the material with such a high rate of corrosion.
The discussion on line 142- 148 about the opaque porous coating and so on, is without evidence.
Author Response
The authors are sincerely grateful to the Reviewer for valuable comments and recommendations. The answers to the comments are in the attached file.

Reviewer 2 Report
The manuscript reports the electrochemical deposition of graphene oxide-Nickel composite film which may have potential applications in anti-corrosion. I think there are several places that can be improved.
- The anti-corrosion performance of the composite film is not that good. So it would be better to domonstrate a better performance. Additionally, from Figure 5, what I have seen is that the GO-Ni film has a higher current, which looks to me that GO-Ni is more easily to be corrosed. Can you explain my confusion?
- SEM-EDX and other characterization is suggested to evaluate the Ni distribution in the film.
- Other parameters should be engineered to optimize the performance of the composite film, such as thickness, Ni% and so on.
Author Response

(The authors gave the same response as above.)

Reviewer 3 Report
Manuscript by Tseluikin et. al, reports the synthesis of Ni-GO composite coating to lower the corrosion rate of Ni coating. The results are scientifically important however, a lot of information about the synthesis and coating process is missing and the results are not presented and explained properly. Additionally, the manuscript is written poorly in terms of English and Grammar. Therefore, I recommend major revision before the manuscript can be reconsidered for further review and publication in Nanomaterials. Please see the detailed comments below -
- Among the various composites of Ni with several carbon-based materials, why Nickel-graphite is of interest? Please explain in the introduction, how the various Ni-Carbon based composites differs?
- I recommend to make a schematic diagram explaining the electrolytic deposition process of Ni-GO, include the pathway of GO and Ni in both voltage sweeps ( from positive to negative and negative to positive voltage sweeps)
- Please correct Figure 1 y-axis and figure 5 both x and y-axis. The commas between the two digits should be decimals, right? Also, all the plots are not consistence, some are images and some are vectors, axis labeling is not consistent. Please format all the figure consistently so that they are easy to understand.
- Similar mistake is done in tables as well. Please format the tables consistently. Include the units with all the results. And, change the comma to decimal in the Table 2, corrosion rate data.
- Please explain the decrease in the potential jump to more electronegative for Ni-GO composites compared to Ni only.
- The GO particles dispersed in the electrolyte are of size 10 micrometer. However, The size in the SEM image (Figure 2) appears to be much bigger. Please explain. Also, is this SEM image of just graphene oxide or Ni-graphene oxide composite?
- Please mention what does the symbol ik, ia, tc, ta, represent? And what are their effects on the deposition process and the coating properties. As the captions of various figures mention that the Ni-GO CEC is obtained at a specific time ratio tc/ta. Please clearly instead their meaning and importance in the deposition process and the properties of the Ni-GO CEC.
- The SEM images in the figure 4 are not clear and of poor quality. Please include better images. Please also include the cross section image of Ni-GO CEC
- Reviewing the manuscript thoroughly, the characterization results (Figure 3 and 4) are mention for the Ni-GO CEC prepared at tc/ta 10:1 , whereas the electrochemical analysis results (Figure 1 and 5) are mentioned for the Ni-GO CEC prepared at tc/ta 16:1. Please be consistent and show both the characterization and electrochemical analysis results for the same kind of CEC.
- Please mention the value of W, A, D for all the time ratio CEC in table 2.
- What do you mean by opaque porous in line 146, page 5? Opacity represents the how much light can transmit through the material. how does that relate to the corrosion?
- Line 150, page 6, “corrosion between the centers” what are these centers? Please clarify. Overall, the hypothesis explaining the low corrosion rate of Ni-GO CEC is not clear. If you think it would be better to explain that with a schematic, please do that. This is the most important part of the manuscript, and needs to be presented clearly.
Author Response

(The authors gave the same response as above.)

Round 2
Reviewer 1 Report
Dear Authors,
Unfortunately, the responses and correction are not enough to qualify for further consideration. SEM data was not improved, the XRD is still wrong analyzed, and the authors only tried to remove the words, but did not improve the discussion. Also, if the authors think that what is available in the literature should not be mentioned again, then the paper is obsolete because of available similar with deeper discussion and thorough characterization.
Reviewer 2 Report
The authors have well revised the manuscript according to the last round review. So it can be accepted.
Reviewer 3 Report
The manuscript has been improved to some extent, however many changes, which are essential for understanding and scientific soundness of the work, have not been incorporated. For example, a schematic diagram explaining how the deposition of Ni is happening in the GO sheets is happening, SEM image of cross section of the Ni-GO composite, consistency in the sample which are characterized and tested for the corrosion performance (samples which are characterized are made at different deposition conditions then the samples which are tested for the performance). Their explanation of the use of different type of samples for characterization and corrosion behavior is not satisfactory because the corrosion testing is done for sample prepared at tc/ta 16:1, which is explained by the characterization result done for sample prepared at tc/ta 10:1, which does not make sense. Therefore, due to the inconsistency in the results and hypothesis, quality of the results, and scientific soundness, I do not recommend this manuscript to be published in the nanomaterials.